# Mechanical Analysis of a Scraping Method to Remove Attached Barnacles

**Chao Li [1], Gang Wang [2],* , Kaiyun Chen [3], Feihong Yun [1] and Liquan Wang [1]**

[1]   College of Mechanical and Electrical Engineering, Harbin Engineering University, Harbin 150001, China; 54529880@hrbeu.edu.cn (C.L.); yunfeihong@hrbeu.edu.cn (F.Y.); wangliquan@hrbeu.edu.cn (L.W.)
[2]   College of Shipbuilding Engineering, Harbin Engineering University, Harbin 150001, China
[3]   College of Mechanical Engineering, Heilongjiang University of Science and Technology, Harbin 150001, China; chenkaiyun@hrbeu.edu.cn
*   Correspondence: wanggang@hrbeu.edu.cn

**Abstract:** In order to clean the marine fouling attached to marine steel piles, a scraping method is proposed in this paper. Barnacles were used to represent a typical object needing removal, in order to estimate the maximum force required in the equipment designed for use in this method. On the basis of the orthogonal cutting theory and the peel zone method, a scraping method and its cutting force model are proposed in this paper for the surface cleaning of marine steel piles. The finite element method was used to verify the analytical model errors. The comparison showed that the relative errors of the cutting force are less than 10%. Our model can be used for cutting force estimation in cleaning equipment design. Our analysis shows that the blade rake angle has a large effect on the cutting force and that the optimum blade rake angle design is a compromise between blade strength and cutting force. We conclude that increasing the blade rake angle can reduce the cutting force in this scraping process; a medium blade rake angle [30°, 60°] is recommended, considering both cutting force and blade strength.

**Keywords:** marine steel pile; barnacle model; orthogonal scraping; mechanical analysis; simulation

## 1. Introduction

In recent decades, a large range of offshore infrastructure has been built to exploit marine resources, including gas and oil. Due to the offshore environment, marine steel piles are subject to the attachment of marine fouling, which results in a huge cost per year to remove them [1]. The removal of marine fouling attached to steel piles is a big challenge that has attracted the interest of both academics and industry. Most marine steel piles are located in the marine splash zone where corrosion is much more severe in other areas [2]. Over time, a rough, thick, hard shell is built-up on the steel pile surface, which causes corrosion. This increases the load of the steel pile, as well as the surface area of the steel pile, which further increases the wave load and current load on the steel pile and results in a shorter life cycle. The corrosion rate in the splash zone is about 3 to 10 times of that in the relatively calm area. At present, anti-corrosion techniques are used extensively, but the steel pile cleaning techniques still need to be investigated to make them cost efficient. Nearly 5000 species of organisms in the ocean are defined as marine fouling [3]. Corrosion is caused directly or indirectly by the attachment, growth, reproduction, metabolism, and death of marine fouling. The organisms grow on the metal surface, produce corrosive substances, or enhance electrochemical corrosion, causing local corrosion such as pitting and crevice corrosion on the surface of the steel structure. In this study, barnacles were used as the most typical and representative specimen to investigate, which play a crucial role in marine fouling damage to marine engineering facilities [4–6].

Commonly, chemical, mechanical, and water jet methods have been used to remove marine fouling. Chemical methods include coating and agent cleaning. The coating method is to cover the surface of steel piles with a layer of biocide and/or coat the pile with an anti-corrosion layer. The chemical method can kill the marine fouling and/or make the adhesion strength lower and thereby ease removal [7–11]. The cleaning method is to spray some acid on the surface [12]. The acid produces not only strong corrosion on both piles and operation devices, such as robots but also pollution in the sea. Mechanical methods include brushing and water jetting. The brushing method is to remove the marine fouling by a rotating brush [13,14]. This method may damage the coating on the original surface and the operation cycle is long. Water jet cleaning methods include high-pressure water jets and cavitation water jets. The high-pressure water jet cleaning method is to use high-pressure water impact to remove marine fouling [15–18]. The high-pressure cavitation water jet method [19–21] is to use the cavitation bubble to remove marine fouling. When the bubble collapses, the energy is released and generates local shear stress in a small area, which acts upon the surface to remove the marine fouling. However, this method involves a long operation time and residues can be seen on the surface. In order to improve the efficiency of the cleaning operation, a new method and device are required, and scraping is the method with most potential.

The scraping method was used by Murakawa and Takeuchi [22] to remove a diamond coating on a substrate, to estimate the adhesion strength by measuring the cutting force. It was also used by Xie et al. [23,24] to measure the bond strength of various metal films on a substrate by applying a dedicated scraper in the measurement. Further, the scraping method was used by Kenji Kaneko [25] to obtain the adhesion strength of the spray coating accurately and improve the reliability of the experiment. In this paper, our scraping method, which combines the orthogonal scraping method and peel zone method to analyze adhesion, is proposed to remove marine fouling on the surface of a steel pile. Our goal is to encourage the use of the scraping method in cleaning system designs, especially as it is suitable for use by robot operations, to take the place of man power in cleaning up marine steel piles. Among the various hard-fouling organisms, barnacles are the most typically used as a model and object for similar fouling remediation studies.

## 2. Barnacle Background and Model Analysis

The barnacle is the main sea creature responsible for fouling. Most barnacles live in the intertidal zone and attach to fixed or floating hard objects in the sea. It takes only a few months for the barnacle larvae (who make a temporary adhesion) to grow into shell-like adults. Barnacles employ a proteinaceous substance for underwater attachment. They secrete a liquid cement to 'glue' themselves to the substrate surface for permanent adhesion. Barnacle cement consists of more than 90% protein and is a multi-protein complex. Barnacle studies have suggested the significance of intermolecular interaction [26]. Four or six harder calcareous shells surround the barnacle from outside, as shown in Figure 1a,b [27]. The barnacle attachment to the substrate is performed by the cement that they secrete from their underside. The cement is sticky and insoluble in water. A cross-section of a barnacle attached to a substrate is shown in Figure 2, with the cement layer scaled in thickness [26].

Jadidi [28] presented a model consisting of a barnacle attached to a cylinder in order to analyze the effect of barnacles on cylindrical vortex-induced vibration. This model was an axisymmetric mound with a trapezoidal cross-section. Based on the previous model, in this research, the barnacle is ideal as it is an axisymmetric mound with a trapezoidal cross-section (calcareous) and has a round cushion (cement) on the bottom, as shown in Figure 3. The cement attaches to a rigid substrate plane.

The geometrical model's bottom diameter is d1, the height average is 12 months, the barnacle is h1 = 5d1/12, the top diameter is d2 = 7d1/12, and the cement cushion height is h2 = d1/24, approximately, as shown in Figure 4.

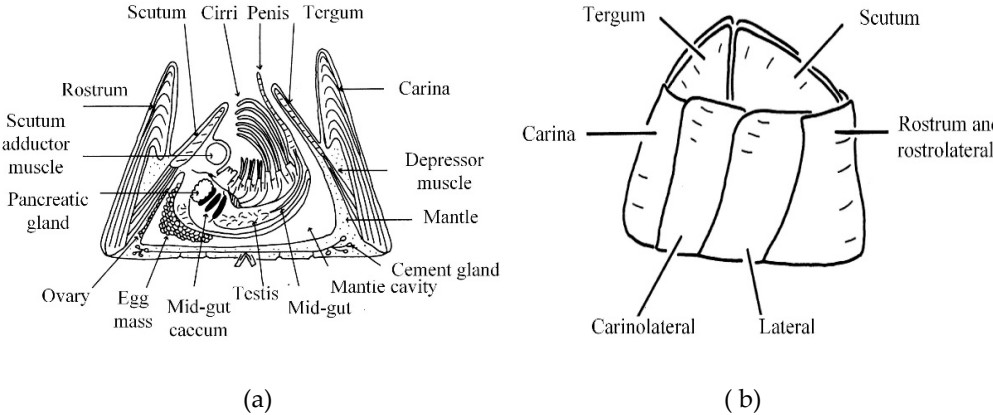

**Figure 1.** Anatomic structure (**a**) and 3D profile (**b**) of barnacle.

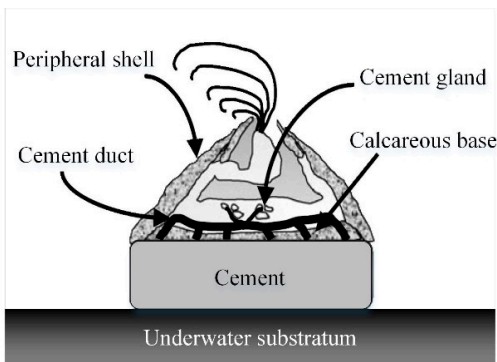

**Figure 2.** The cross-section of a barnacle on a foreign substratum.

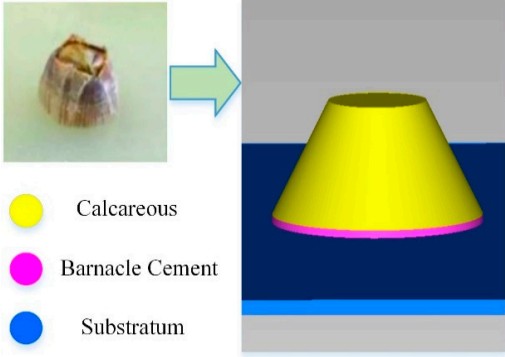

**Figure 3.** The cross-section of a barnacle on a foreign substratum.

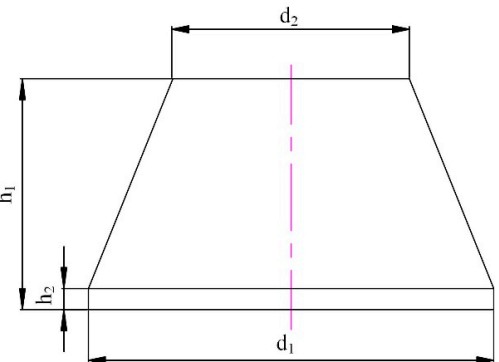

**Figure 4.** Barnacle geometrical model and dimensions.

The size/diameter of different kinds of barnacles varies from 8 to 43 mm [29]. Most of our information on barnacle growth relates to the *Balanomorpha genus*, and in particular, to *Semibalanus balanoides* (Previous name *Balanus balanoides*). *S. balanoides* is a sessile barnacle that attaches to hard substrates and is a widely-distributed intertidal species found on both sides of the North Atlantic and the eastern North Pacific [30,31]. This species has a highly synchronized settlement; therefore, a single cohort can be readily followed. It is relatively easy to measure the diameter and it has specific representativeness. Its adhesion force increases monotonously with the area and age of the *S. balanoides* attached, and the attachment area increases with time [32], as shown in Figure 5.

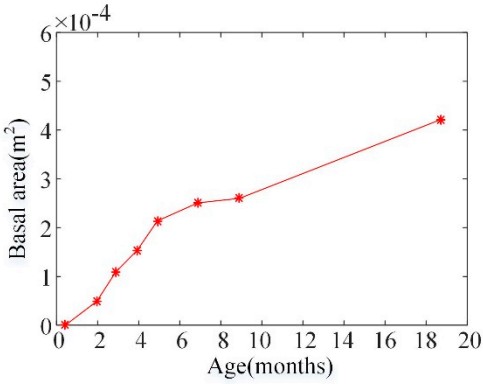

**Figure 5.** Bottom area of *S. balanoides* with age.

## 3. Removing Process and Methods

### 3.1. Analysis of Removing Process

The barnacle was attached to the rigid substrate by cement. It was hard to remove it because the calcium shells strongly attached to the cement. In this paper, a scraping method was proposed to analyze the force required for barnacle cleaning, which combined the orthogonal cutting method and the peel zone method proposed by Pesika [33,34]. The process of scraping the barnacle was organized into four stages, as shown in Figure 6.

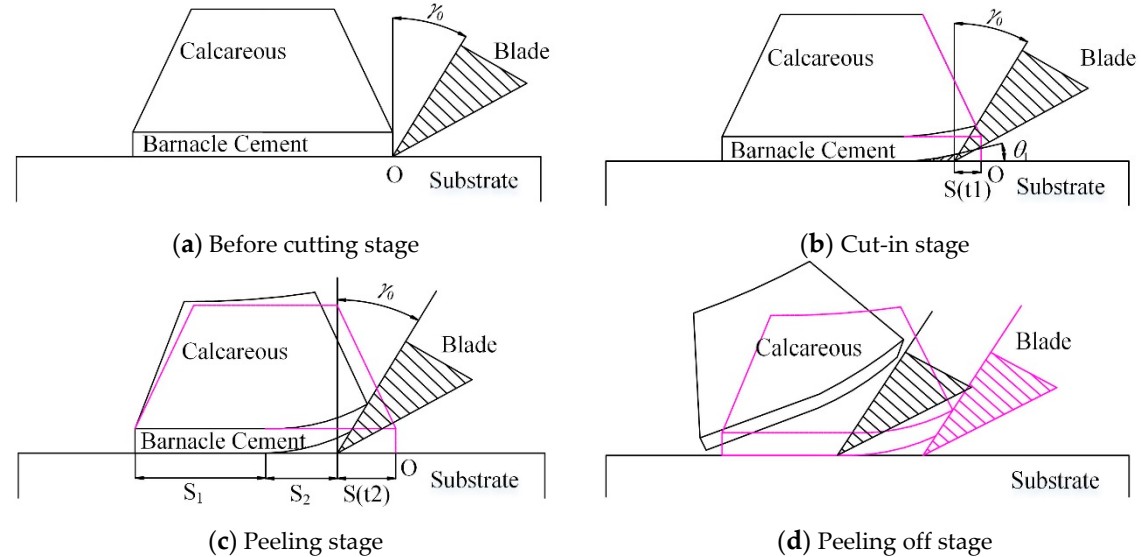

**Figure 6.** Schematic diagram of cleaning the barnacle process.

The first stage was the before-cutting stage. The blade with a rake angle of $\gamma_0$ moved towards the barnacle at a certain speed until it engaged with the barnacle cement at point O.

The second stage was the cut-in stage. The blade cut into the cement from point O until the blade touched the calcareous shell at the moment of t1, with a displacement of S (t1), and a peel angle $\theta_1$ was produced.

The third stage was the peeling stage. The blade further moved to S(t2) at the moment of t2. The blade represented the normal and tangential forces of the shell. The normal force applied to the rake face was the main force that was applied to the barnacle—making the calcareous shell deform, gradually lift, and almost break away from the substrate. The remaining adhesion length was $S_1$. The curved portion forms of the peel zone, with the corresponding length of $S_2$, and $d_1 = S_1 + S_2 + S(t2)$, had a range of peel angles at $0° < \theta \leq 90°$.

The fourth stage was the peeling-off stage. The total adhesion of the barnacle decreased, in addition to the force on the blade, as the blade kept moving until the barnacle was fully peeled away from the substrate.

### 3.2. Force Analysis

Scraping the barnacle that was attached to the substrate was, in some ways, similar to peeling-off the gecko spatula pad. Barnacle cement and gecko spatula pads are both viscoelastic materials that were attached to the substrate. However, the calcareous shell of the barnacle restricted the large deformation of the cement when the barnacle was peeled-off, which was the difference from peeling-off the gecko spatula pad. The calcareous shell deformed and lifted up after coming into contact with the blade's surface. The peel zone could be formed in a similar way to the spatula pad peeling. The peel zone was the area where the cement was peeled-off from the substrate and some of the adhesive fibers still remained, as shown in Figure 7. The area enclosed by points $x_1$, $x_2$ and the tool-tip point M was the peeling area, corresponding to the horizontal distance $S_2$. The vertical distance from $x_2$ to the surface of the substrate was D, which was the critical distance; the adhesion failed at any point when the vertical distance to the substrate surface was larger than D. The other variables are defined in the next section. The adhesion force $F_A$ on the substrate could be analyzed and calculated according to the adhesion mechanism of the cement, and the adhesion force of the peel zone, $F_{PZ}$, could be analyzed by Pesika's peel zone method.

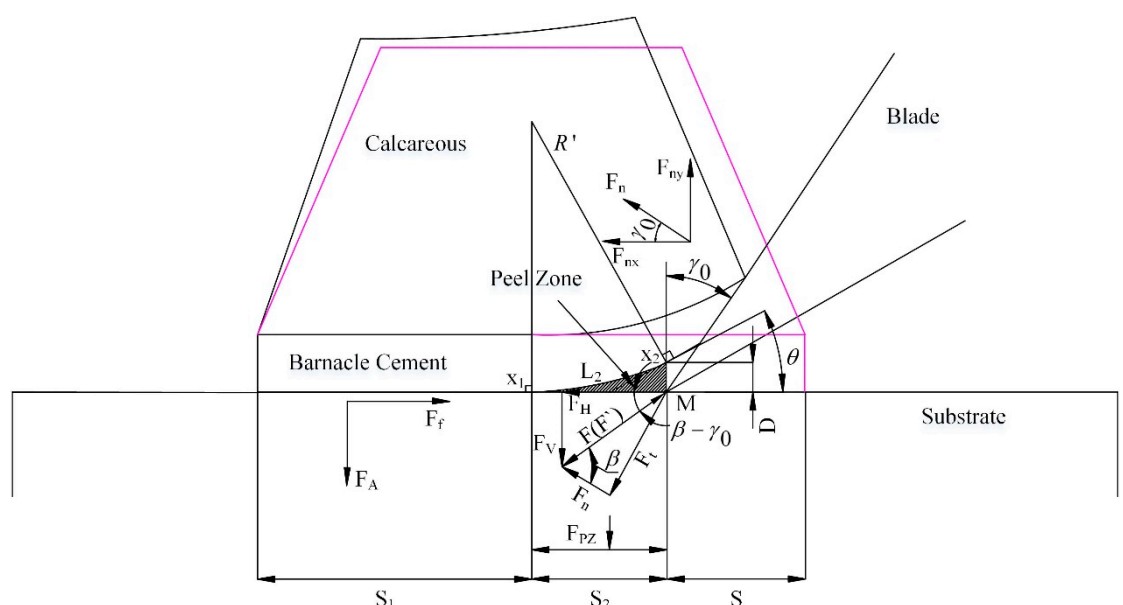

**Figure 7.** Detailed analysis of the barnacle.

### 3.3. Force on Blade

The orthogonal cutting theory used the calculation from the second stage of the scraping. We can assume that the cutting edge of the blade was absolutely sharp, the flank face was not in contact with the substrate, and the friction of the flank face was zero [35,36]. Two forces were applied to the calcareous shell, as shown in Figure 7: the normal force $F_n$ and the tangential friction force $F_t$, which are from the rake face and F is the resulting force. F′ was the counterforce of F and was equal in amplitude and opposite in direction. The counterforce F′ could be decomposed in the horizontal direction, $F_H$, and the vertical direction, $F_V$. β was the angle between F and $F_n$, and β-$\gamma_0$ was the angle between F′ and $F_H$.

The calcareous shell was deformed by the normal force $F_n$ of the rake face and $F_n$ could be decomposed into normal force $F_{nx}$ and tangential force $F_{ny}$. The following relationship applies:

$$F_{ny} = \frac{F_H \cos \beta \sin \gamma_0}{\cos(\beta - \gamma_0)},\tag{1}$$

where β is also called the friction angle, and $\gamma_0$ is the fake angle of the blade.

### 3.4. Adhesion Force in the Peel Zone

In the second stage, the barnacled bottom, with a radius of R, was composed of three zones: the divided zone, the peeling zone, and the adhesion zone, as shown in Figure 8. The pink shadow part was the adhesion zone with a length of $S_1$, a normal adhesion force $F_A$ and a horizontal friction force $F_f$. The blue shadow was the peel zone with a length of $S_2$ and an adhesion force $F_{PZ}$. The amplitudes of $F_A$ and $F_{PZ}$ could be calculated individually.

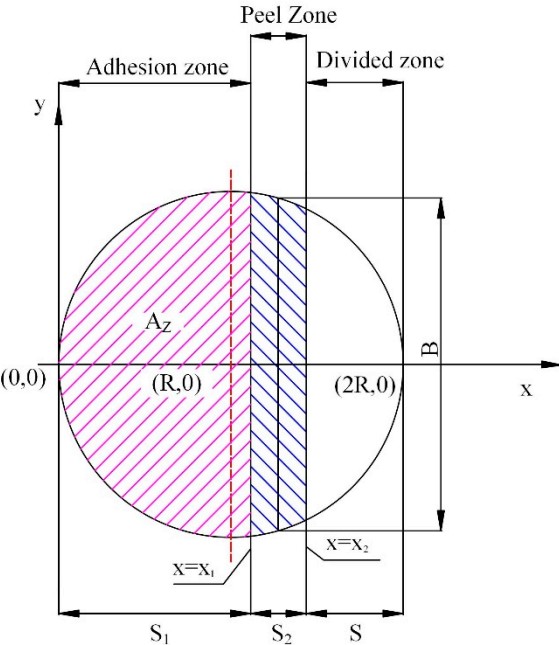

**Figure 8.** Zones definitions in the second stage of the barnacle scraping.

The calcareous shell was deformed by the normal force of $F_n$ on the rake face. Suppose that the bottom deformation of the shell is an arc with a radius of R′ and the cement deformation follows the calcium shell. When the cement was peeled off, the bottom surface kept the same arc and the arc length between $x_1$ and $x_2$ was $L_2$. When θ was small, $L_2 = R'\theta \approx S_2$. The cement volume in the peel zone was enclosed by three points—$x_1$, $x_2$ and M. $x_1$ is the contact point where the cement started to separate away from the substrate. $x_2$ was the point at which the last fiber of cement was broken on the substrate.

M was the blade tip. The angle between $x_1$ and $x_2$ was the peel angle, $\theta$. According to the empirical formula, the relationship between R' and $\theta$ was R' = 4215 × $\theta^{-1.35}$ [34]. The width B in the peel zone between $x_1$ and $x_2$ was B = $2\sqrt{R^2 - (R-S)^2}$, as shown in Figure 8. The adhesion force of peel zone is:

$$F_{PZ} = \frac{A\sqrt{R'\left(R^2 - (R-S)^2\right)}}{8\sqrt{2}D^{\frac{5}{2}}}, \tag{2}$$

where A is the Hamaker constant, D is the surface gap.

In Reference [33], the "gecko spatula pad is composed of β-keratin" and is a kind of protein. In Reference [26], "more than 90% of barnacle cement is composed of proteins and is a multi-protein complex". In References [26,33,34], their adhesion mechanisms were all caused by intermolecular interactions, and the thickness of the gecko spatula pad and barnacle cement were of the same order of magnitude. Based on the above similarities, we also made assumptions that ensured that the empirical formulas were used to estimate the adhesion of the peel zone.

When S = 0.024 m, the relationship between the peeling angle and the adhesion force was as shown in Figure 9. The adhesive force decreased quickly when 0° < $\theta$ < 10° and gently when $\theta$ > 20°.

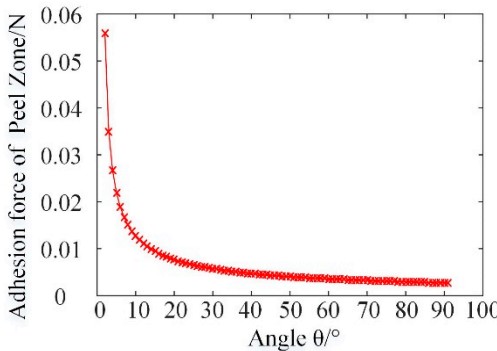

**Figure 9.** The adhesion force of the peel zone at different angles $\theta$.

When the blade moved from 0 to 0.024 m, the adhesive force in the peel zone, with a peel angle from 1° to 10°, was as shown in Figure 10. When S = 0.012 m, the adhesion force of the peel zone reached its maximum, and gradually reduced after that because its bottom projection was circular, as shown in Figure 8. As is known from Equation (2), the adhesion force of the peel zone was affected by the width B in the removal process; B first increased and then decreased, and reached the maximum at S = 0.012 m.

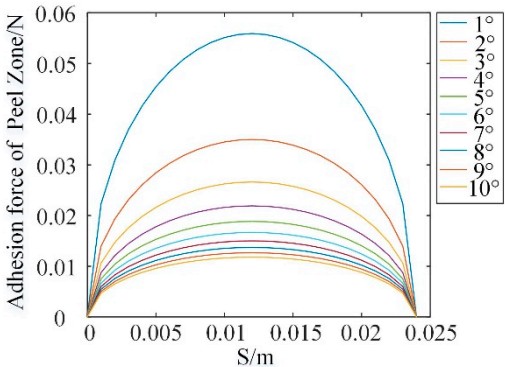

**Figure 10.** Adhesion force of the peel zone vs blade displacement, S, with peel angle from 1° to 10°.

The area of the adhesion zone, $A_Z$, as shown in Figure 8, is:

$$A_Z = 2 \int_0^{x_1} \sqrt{R^2 - (X-R)^2} dX, \tag{3}$$

where R = 0.012 m, $A_Z$ is expressed as:

$$A_Z = \frac{9\pi}{125000} + \frac{9\arcsin\left(\frac{250x_1}{3} - 1\right)}{62500} + \sqrt{\frac{9}{62500} - \left(x_1 - \frac{3}{250}\right)^2} \times \left(x_1 - \frac{3}{250}\right), \tag{4}$$

where $S_1 = d_1 - S_2 - S$ and $S_2 \approx R'\theta$. The adhesion strength of the adult barnacle was P = $9.252 \times 10^5$ N/m$^2$ [26,32]. Suppose that the barnacles to be removed were more than one year old, in which case the remaining adhesion force $F_A$ can be expressed as follows:

$$F_A = P \times \left( \frac{9\pi}{125000} + \frac{9\arcsin\left(\frac{250(2R - R'\theta - S)}{3} - 1\right)}{62500} + \sqrt{\frac{9}{62500} - \left((2R - R'\theta - S) - \frac{3}{250}\right)^2} \times \left((2R - R'\theta - S) - \frac{3}{250}\right) \right), \tag{5}$$

In the vertical direction, the following equation applies:

$$F_{ny} = F_A + F_{PZ}, \tag{6}$$

Substitute Equations (2) and (5) into (6) and, $F_{ny}$ can be rewritten as follows:

$$F_{ny} = \left[ P \times \left( \frac{9\pi}{125000} + \frac{9\arcsin\left(\frac{250(2R - R'\theta - S)}{3} - 1\right)}{62500} + \right. \right.$$
$$\left. \left. \sqrt{\frac{9}{62500} - \left((2R - R'\theta - S) - \frac{3}{250}\right)^2} \times \left((2R - R'\theta - S) - \frac{3}{250}\right) \right) \right] + \left( \frac{A \cdot 2\sqrt{R^2 - (R-S)^2} \cdot (R')^{\frac{1}{2}}}{16\sqrt{2}D^{\frac{5}{2}}} \right), \tag{7}$$

The adhesive force, with regard to the blade displacement, was as shown in Figure 11. The adhesive force was the maximum at the zero point where the blade did not cut in, and when the blade kept moving, the adhesion force decreased due to the adhesion area decreasing. When the blade displacement equaled the barnacle diameter, 2R, the adhesion force was 0 and the barnacle was fully removed.

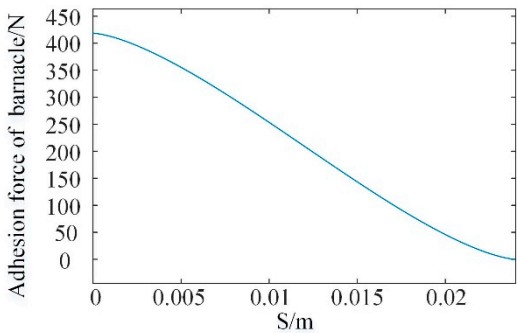

**Figure 11.** Adhesion force vs blade displacement with barnacle diameter $d_1$ = 0.24 mm.

Substitute Equations (1) into (7) and, $F_H$ can be expressed as:

$$F_H = \frac{\cos(\beta - \gamma_0)}{\cos\beta\sin\gamma_0}\left\{\left[P\times\left(\frac{9\pi}{125000} + \frac{9\arcsin\left(\frac{250(2R - R'\theta - S)}{3} - 1\right)}{62500} + \right.\right.\right.$$
$$\left.\left.\left.\sqrt{\frac{9}{62500} - \left((2R - R'\theta - S) - \frac{3}{250}\right)^2}\times\left((2R - R'\theta - S) - \frac{3}{250}\right)\right)\right] + \left(\frac{A\cdot 2\sqrt{R^2 - (R - S)^2}\cdot(R')^{\frac{1}{2}}}{16\sqrt{2}D^{\frac{5}{2}}}\right)\right\} \tag{8}$$

where $A = 0.4 \times 10^{-19}$ and $R = 0.012$ m. The maximum friction force corresponded to the peel angle, $\theta = 10°$, and $R' = 4215 \times \theta^{-1.35}$ [34]. The friction coefficient of the cement on the substrate and the rake face of the blade was 0.25 [33], which corresponded to $\beta = 14°$. The cutting force $F_H$, with respect to the blade displacement and the rake angle from 10° to 90°, is as shown in Figure 12.

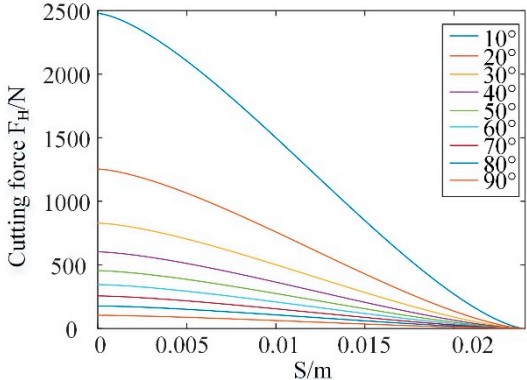

**Figure 12.** The cutting force Fc with the blade displacement from 0 to 0.024 m and the rake angle from 10° to 90°.

The rake angle affects the cutting force $F_H$ and can also be plotted as a 3D surface, as shown in Figure 13. The blade rake angle had a significant effect on the cutting force. Reducing the rake angle could increase the cutting force considerably in the range of (0°–45°) and this trend was relatively gentle in the rake angle range of (45°–90°). The rake angle could be a compromise between the cutting force and blade strength/life cycle. The blade rake angle should be a medium one, such as [30°, 60°].

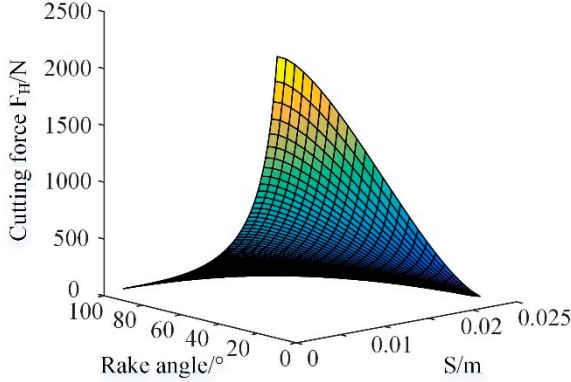

**Figure 13.** 3D surface of the cutting force $F_H$ with the blade displacement from 0 to 0.024 m and the rake angle from 10° to 90°.

$F_V$ can be obtained with the same method:

$$F_V = \frac{\sin(\beta - \gamma_0)}{\cos\beta\sin\gamma_0}\left\{\left[P\times\left(\frac{9\pi}{125000} + \frac{9\arcsin\left(\frac{250(2R - R'\theta - S)}{3} - 1\right)}{62500} + \right.\right.\right.$$
$$\left.\left.\left.\sqrt{\frac{9}{62500} - \left((2R - R'\theta - S) - \frac{3}{250}\right)^2}\times\left((2R - R'\theta - S) - \frac{3}{250}\right)\right)\right] + \left(\frac{A\cdot 2\sqrt{R^2 - (R - S)^2}\cdot(R')^{\frac{1}{2}}}{16\sqrt{2}D^{\frac{5}{2}}}\right)\right\} \quad (9)$$

The cutting forces $F_H$ and $F_V$, with a blade rake angle of 60°, are plotted in Figure 14. The absolutes of both $F_H$ and $F_V$ decreased as the blade kept moving until $S = 0.024$ m, where both cutting forces were 0 and the barnacle was fully removed. The adhesion force in the vertical direction was $F_{ny} = F_A + F_{PZ} = 419$ N and the maximum horizontal cutting force was $F_H = 346$ N.

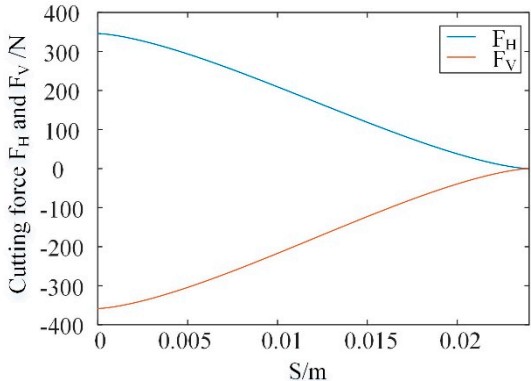

**Figure 14.** The 60° rake angle of the blade corresponded to the cutting force $F_H$ and $F_V$.

## 4. Finite Element Analysis and Results

To verify the analytical model, 3D finite element models are created in Ls-dyna with Hypermesh, as shown in Figure 15. The barnacle size, $d_1$, is 0.024 m in diameter. The height, cement thickness, and top diameter are calculated according to Figure 4. The blade rake angle was 60°. The blade velocity is 0.7 m/s. The solid element is used for the blade, barnacle shell and cement, and the substrate is set as the shell elements to minimize the computation load.

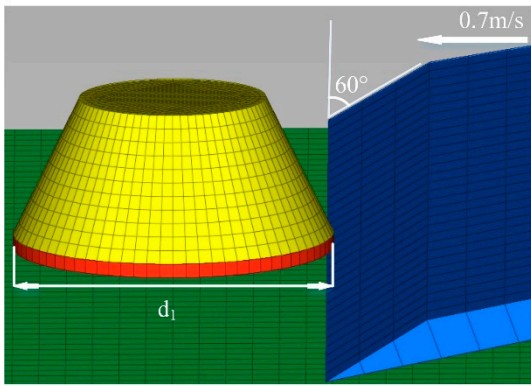

**Figure 15.** The simulation diagram of barnacle.

The material of the blade and substrate is elastic, the density is $\varrho = 7.93 \times 10^3$ kg/m³, the bulk modulus of elasticity is $K = 1.95 \times 10^5$ MPa, and the Poisson's ratio is $\gamma = 0.247$. Calcareous shell is a plastic material, the density is $\varrho = 2.6 \times 10^3$ kg/m³, the bulk modulus of elasticity is $K = 5 \times 10^4$ Mpa,

and the Poisson's ratio is $\gamma = 0.3$; the material of the cement is viscoelastic, the density is $\varrho = 1.19 \times 10^3$ kg/m$^3$, and the bulk modulus of elasticity is K = 100 MPa. The model has meshed with 12,162 elements and 15,405 nodes in total.

The contact with a failure, CONTACT_AUTOMATIC_SURFACE_TO_SURFACE_TIEBREAK, is set between the cement and substrate; the contact (no failure), CONTACT_AUTOMATIC_ SURFACE_ TO_SURFACE, is set between the calcareous shell, cement, and blade. In total, four contact groups are set in this simulation. The friction between the flank face and the substrate is set as zero. The bottom substrate is set as fixed in six freedom degrees and the blade is set as -0.7 m/s in the x direction.

The adhesion forces of both simulation and analytical results are plotted in Figure 16 with respect to the blade displacement.

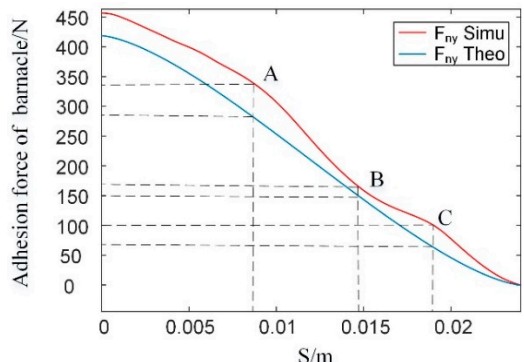

**Figure 16.** Simulation and analytical results of barnacle adhesion force.

The trend of the analytical result from the adhesion force agrees with the trend exhibited for the simulation; the analytical result is a little lower than the simulation result in the full range of the blade displacement, as shown in Figure 16. The maximum and minimum errors occur at point A and B, with the maximum relative error of 14.7% at point A and the minimum relative error of 11.8% at point B. At point C, the relative error is 13.5%, which is between the maximum and the minimum values.

The analytical and simulation results of the cutting forces, $F_H$ and $F_V$, are plotted in Figure 17. The trends of analytical results agree with the simulations. The analytical and simulation curves of the cutting force, $F_H$, intersect at points D, E and F. The relative errors between them are less than 10%. The analytical and simulation results of the cutting force, $F_V$, are in complete agreement at points of G and H. The maximum error occurs at the start point where S = 0. The error between the simulation and analytical results is the result of the assumption of the Peel Zone method, particularly the assumption of the arc section of the cement in the peel zone and the radius, R', being considered as not being a constant but a function of S.

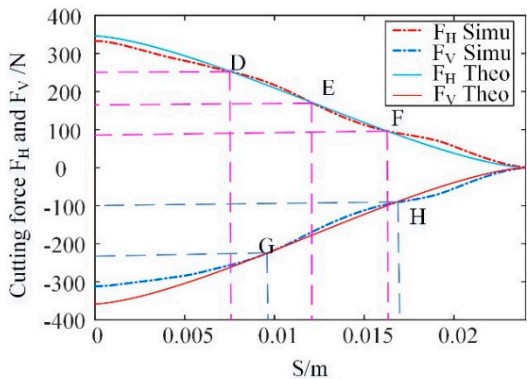

**Figure 17.** Analytical and simulation results of the cutting forces $F_H$ and $F_V$.

### 5. Conclusions

In this paper, on the basis of the orthogonal cutting theory and the peel zone method, an analytical model of the cutting forces is proposed to scrape barnacles with blades. On the basis of this analysis, the following conclusions can be drawn:

(1)　The analytical model is verified with a finite element simulation and the comparison shows the relative error of this model to be less than 10%. The model can be used to estimate the cutting force in barnacle scraping operation in order to support related equipment design.

(2)　The blade rake angle has a significant effect on the cutting force. Reducing the rake angle can increase the cutting force considerably in the range of (0°–45°) and this trend is relatively gentle in the rake angle range of (45°–90°). The rake angle can be a compromise between cutting force and blade strength/life cycle. The conclusion can be drawn that the larger the blade rake angle, the smaller the cutting force required to remove the attached barnacle. The blade rake angle should be a medium one, such as [30°, 60°].

Further research will be carried out in the following areas: (1) barnacle size/age effects on the cutting force, (2) barnacle size/age-dependent material properties for the cement and shell, (3) rake angle and barnacle size/age effects on the peel angle, and (4) the error correction that poses a challenge to minimizing the model errors and coefficients, which can be introduced to correct the cutting forces and adhesion force with respect of the barnacle age/size.

**Author Contributions:** Formal analysis, C.L. and G.W.; Methodology, C.L.; Writing—original draft preparation, C.L., and G.W.; Writing—review and editing, K.C., F.Y. and L.W., Validation, K.C., L.W. and F.Y.; Supervision L.W.; Funding acquisition, G.W. and F.Y. All authors have read and agreed to the published version of the manuscript.

**Funding:** This research was funded by National Key Research and Development Project [Grant No. 2018YFF01012900], National Natural Science Foundation of China [Grant No. 51779059, 51409058, 61633009], Fundamental Research Funds for the Central Universities [Grant No. 3072019CF0705], China Postdoctoral Science Foundation [Grant No. 2018M630343], and Science and Technology on Underwater Vehicle Laboratory [Grant No. 614221503041701].

**Acknowledgments:** The authors gratefully acknowledge the financial support from National Key Research and Development Project [Grant No. 2018YFF01012900], National Natural Science Foundation of China [Grant No. 51779059, 51409058, 61633009], Fundamental Research Funds for the Central Universities [Grant No. 3072019CF0705], China Postdoctoral Science Foundation [Grant No. 2018M630343], and Science and Technology on Underwater Vehicle Laboratory [Grant No. 614221503041701].

**Conflicts of Interest:** The authors declare no conflict of interest.

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
