# Peer review of "Mechanical Analysis of a Scraping Method to Remove Attached Barnacles"

_jmse, doi:10.3390/jmse8030150_

Round 1

Reviewer 1 Report

Dear Authors,

The present study deals with the evaluation of a scraping method for removing marine fouling on steel pile surface. In order to improve the quality of the manuscript, I suggest some changes regarding language, spelling and organization of content, as it follows:

Please rephrase the first and last sentence of the abstract, it is not properly stated Lines 27-28: please remove repetitive word “fouling” and “marine fouling” Line 30: “corrosion is much more severe than IN other areas” Lines 35-36: please correct or explain the sentence “Nearly 5,000 species of organisms in the ocean are classified as marine fouling which mentioned above.” Line 40: please explain sentence “In this paper, barnacles are used as the object to investigate” Lines 54-55: “However, this method is not so efficient and no guaranty for the fully removal of the marine fouling.” - this not a proper way to express disadvantages of different methods Line 55-56: Please rephrase “The above research is the existing method for removing marine fouling” in order to refer to all of the above-mentioned scraping methods Lines 59-63: please properly introduce the state of the art and make connection with the before-mentioned sraping methods Lines 63-68: please properly explain the aim of the study, it is not clear Line 87: please explain that the previous study represents the starting point of your study Line 98: Balanomorpha genus? Lines 99-101: please rephrase “ balanoides is a widely-distributed intertidal species found on both sides of the North Atlantic and the eastern North Pacific [30, 31], which is a sessile barnacle attached to hard substrates.” Lines 109-110: please rephrase “It is difficult to be removed as the hard calcium shells growing on the cement and becoming more and more firm as the time passing.” Lines 119-133: no need to mention each time “as shown in Figure 6 A-D” Line 136: please explain the word “they” in “They are all the viscoelastic material” Line 138: please explain the difference compared to what?

Please perform the suggested changes in order to improve the structure and quality of the presented information.

Reviewer 2 Report

In the last review I asked you why you used for the bernacle an empirical formula that is valid for gecko spatula (i.e. R = 4215xtheta^(-1.35)). You reply me properly with references that demonstrate the validity of the assumption.  Why you do not put these commenst also in the paper?

Anyway, in my opinion the paper can be accepted in this form.

Author Response

The authors would like to thank the reviewer for useful feedback, and the manuscript has been updated accordingly. The response to the revision is as follows:

Point 1: In the last review I asked you why you used for the bernacle an empirical formula that is valid for gecko spatula (i.e. R = 4215xtheta^(-1.35)). You reply me properly with references that demonstrate the validity of the assumption.  Why you do not put these commenst also in the paper?

Response 1: Lines 178-183, we added “In references [33], “gecko spatula pad is composed of β-keratin”, which is a kind of protein. In references [26], “more than 90% of barnacle cement is composewd of proteins and is a multi-protein complex”. In the references [26] [33] [34] that their adhesion mechanisms are all caused by intermolecular interactions, the thickness of gecko spatula pad and barnacle cement are on the same order of magnitude. Based on the above similarities, we also make assumptions which ensure the empirical formulas are used to estimate the adhesion of Peel Zone.” to explain why we use an empirical formula that is valid for gecko spatula. It can be seen from the manuscript references [33], “gecko spatula pad is composed of β-keratin”, which is a kind of protein. It also can be seen from the references [26], “more than 90% of barnacle cement is composed of proteins and is a multi-protein complex”. It can be seen in the references [26] [33] [34] that their adhesion mechanisms are all caused by intermolecular interactions, the thickness of gecko spatula pad and barnacle cement are on the same order of magnitude. In references [33] [34] They made the following assumptions: “We assume that the curvature of the tape backing is circular and that the length of the peel zone on the surface is equal to the arc length of the tape backing up to the point of the last fibril or filament.”, this assumption ensures the feasibility of empirical formula. Based on the above similarities, we also make assumptions: “Suppose that the bottom deformation of the shell is an arc with a radius of R` and the cement deformation follows the calcium shell. When the cement is peeled off, the bottom surface keeps the same arc and the arc length between x1 and x2 is L2. When θ is small, L2= R`θ≈S2.”, which ensure the use of empirical formulas are used to initially estimate the adhesion of Peel Zone.